# Alternate Roles of Sox Transcription Factors beyond Transcription Initiation

**DOI:** 10.3390/ijms22115949

**Published:** 2021-05-31

**Authors:** Yuli Zhang, Linlin Hou

**Affiliations:** Department of Biochemistry, Molecular Cancer Research Center, School of Medicine, Sun Yat-Sen University, Shenzhen 518107, China; zhangyuli970909@163.com

**Keywords:** Sox proteins, transcription factor, transcription, co-transcriptional splicing, translational control

## Abstract

Sox proteins are known as crucial transcription factors for many developmental processes and for a wide range of common diseases. They were believed to specifically bind and bend DNA with other transcription factors and elicit transcriptional activation or repression activities in the early stage of transcription. However, their functions are not limited to transcription initiation. It has been showed that Sox proteins are involved in the regulation of alternative splicing regulatory networks and translational control. In this review, we discuss the current knowledge on how Sox transcription factors such as Sox2, Sry, Sox6, and Sox9 allow the coordination of co-transcriptional splicing and also the mechanism of SOX4-mediated translational control in the context of RNA polymerase III.

## 1. Introduction

Gene expression is a process that transforms genomic information into biological functions. Gene expression in eukaryotes can be regulated at multiple levels. Though each individual level can be studied in isolation in vitro, it becomes clear that they are intimately linked within cells. Transcription factors (TFs) are a large class of DNA-binding proteins that play a key role in gene transcription. Nevertheless, increasing evidence shows that they can function as cross-talkers in several levels of gene expression. 

Sox TFs exist in all metazoans and regulate many developmental processes and homeostasis in adult tissues. The 20 Sox proteins are characterized by a highly conserved Sry-related high-mobility-group (HMG) box. Depending on the sequence similarities of the HMG box, Sox TFs are classified into 9 groups (A, B1, B2, C, D, E, F, G, H) [1,2]. The HMG box folds into an L-shape consisting of three α-helices and two flexible N- and C-terminal extensions, and sequence-specifically binds with a C_1_T_2_T_3_T_4_G_5_T_6_C_7_-like motif. All bases of the core of the DNA motif are directly bound by amino acids in base-specific interactions, and amino acids engaged in base-specific DNA interactions are constant amongst all Sox TFs. The HMG box interacts with the minor groove of the DNA duplex and causes an overall bend of 60–70° towards the major grove, which may in turn regulate the functions of Sox proteins [1]. Notably, Sox TFs display their gene regulatory functions only by forming complexes with partner factors [3,4]. The partner factors can be homologous or heterologous. For example, SoxB1 TFs co-work with heterologous partner factors belonging to TF families such as the PAX and POU families [1]; SoxD TFs form homodimer via their coiled-coil motifs; SoxE TFs can employ both homologous and heterologous partners [3].

Outside the HMG box, strong homology of amino acid sequences and overall organization of protein domains are found only within a group, although this does not mean that synonymous function only exists within each group. Sox2 (a member of the SoxB1 group) and Sox9 (a member of the SoxE group) are proposed to engage at regions exposed on nucleosomes and are defined as pioneer factors [5,6,7]. Recent studies showed that Sry (a member of the SoxA group), SOX2, SOX6 (a member of the SoxD group), and Sox9 play roles in pre-mRNA alternative splicing [8,9,10,11]. A growing number of studies in recent years have shown that that the non-HMG box regions, especially the C-terminal transactivation domain, can act as recruitment platforms connecting Sox TFs with their partner factors such as TFs, splicing factors, and RNAs, suggesting the regulation of Sox TFs in many gene expression steps. In this review, we focus on the Sox TFs that have alternate roles in the regulation of gene expression beyond the early stage of transcription, refine the understanding of ‘transactivation’, and provide new insights into the mechanisms of cell fate decision and development.

## 2. Roles of Sox TFs in Splicing

Eukaryotes possess three distinguishing nuclear RNA polymerases that target different classes of genes. RNA polymerase I (Pol I) specifically aims to transcribe large ribosomal RNA (rRNA) genes, including 28S, 5.8S, and 18S cytoplasmic rRNA. RNA polymerase III (Pol III) is devoted to the transcription of abundant small RNAs containing 5S rRNA, all transfer RNAs (tRNAs), and some small RNAs involved in pre-mRNA splicing and protein transport that are not transcribed by Pol II [12]. Pol II mainly targets all protein-coding genes and plenty of non-coding RNAs in eukaryotic genomes. In general, TFs selectively bind to the DNA regulatory elements located upstream or downstream of the target genes, recruit transcription initiation machinery, which is composed of Pol II and other partner components, and orchestrate gene expression programs [1]. Transcription by Pol II requires the assembly of a preinitiation complex (PIC) composed of general TFs (GTFs, e.g., TFIIA, TFIIB, TFIID, TFIIE, TFIIF, TFIIH) and mediators bound at the promoter. The constituent GTFs are not one-size-fits-all. They are distinct in the transcription of different types of genes or in different cell types [13,14]. Once bound, the PIC opens the promoter DNA and initiates transcription. Notably, recent studies show that Pol II transcribes both unidirectionally and bidirectionally. TFIID is a core promoter recognition factor consisting of TATA box binding protein TBP and 14 TBP-associated factors (TAFs) [15]. TBP binds to TATA or TATA-like box upstream of the transcription start site (TSS) in a single orientation and induces a ~90° bend of DNA helix [16,17]. The TBP-DNA complex is further stabilized by GTFs TFIIA and TFIIB, forming an interaction network within the PIC [18,19,20]. The binding of TBP to the TATA box is believed to orient the PIC on the promoter and drive unidirectional transcription [21]. However, in mammals, ~80% of active promoters display bidirectional transcription initiation [22,23]. Promoters involved in such divergent transcription are typically TATA-less sequences such as the Inr element, the downstream promoter element (DPE), X core promoter element 2 (XCPE2), and CpG islands [24,25,26,27]. TATA-less promoters, e.g., those abundant of CpG islands, have no intrinsic preference for TFIID. Certain activators such as TF Spl, which binds unmethylated CpG islands, are proposed to efficiently recruit the TFIID complex to the TATA-less promoter via a tethering factor that physically interacts with TBP and initiate the bidirectional transcription [28,29]. Intriguingly, transcription initiation also happens at enhancers. PIC and specific TFs bind to many distal enhancers in a combinatorial mode, display bidirectional transcription initiation, and generate short transcripts (eRNA) that will further mediate the activation of target genes [30,31,32]. In contrast to the GTFs that bind to the core promoter region to facilitate the binding of Pol II, sequence-specific DNA-binding TFs, such as Sox2 (interact with TFIID) and Brn2 (interact with TBP), bind to regions outside of the core promoter region and directly or indirectly associate with the factors at the core promoter to stimulate or repress the activity of Pol II [33,34]. 

The splicing of pre-mRNAs in eukaryotes is tightly coordinated with transcription. Nascent RNAs are mostly spliced as soon as Pol II is downstream of introns during transcription elongation [35,36]. This process is also termed co-transcriptional splicing. A ribonucleoprotein complex known as the spliceosome assembles surrounding the intron/exon junctions and alternatively splices pre-mRNAs in association with different auxiliary components. Approximately 40–60% of human genes generate multiple mRNA isoforms from one single pre-mRNA, and up to 88% of the alternative splicing isoforms vary the protein products [37,38]. Hence, splicing is also a crucial step that controls gene expression. Several Sox TFs have been found to carry out co-transcriptional regulation of gene expression in the context of Pol II-mediated transcription (Table 1).

### 2.1. Sox2

Sox2 is a determinant in the maintenance of embryonic and neural stem cells. It presents diverse regulatory roles, especially in chromatin organization and transcriptional regulation. In addition to the general function as a TF, there is strong evidence for its role as a pioneer factor that unwraps nucleosomal DNA to enable gene expression from areas of highly packed chromatin. Cryo-electron microscopy (cryo-EM) structure and SeEN-seq data show that the HMG domain of Sox2 binds to nucleosomal entry-exit sites [39,40]. Together with Oct4, Sox2 kinks the DNA ~90° away from the histone octamer, initiates chromatin opening using the binding energy, which increases the accessibility of nucleosomal DNA, and thereby facilitates subsequent transcription [39,40]. TFs such as Klf4, Nanog, Brn2, Pax6, and ATP-consuming chromatin remodelers such as Chd7 cooperate with Sox2 for efficient chromatin opening [41,42,43,44]. Histone modifications can modulate the accessibility of chromatin. Some histone modifications have been identified to contribute to human embryonic cells’ (hESCs’) fate determination through alternative splicing regulation [45], revealing that Sox2 may be a coordinator involved. 

Ng et al. found that long noncoding RNA ES1 (lncRNA_ES1) and long noncoding RNA ES2 (lncRNA_ES2) physically interact with Sox2 in hESCs and were suggested to maintain pluripotency in a Sox2-dependent mode [46]. In addition, Sox2 and lncRNA RMST co-exist in a transcriptional complex during neuronal differentiation and they regulate a group of downstream gene targets that play important roles for this process [47]. Yet, these studies did not decipher whether Sox2 interacts with RNA directly or through Sox2-associating proteins that have RNA-binding property. 

Lately, Sox2 was identified as an RNA-binding protein that contains no classic RNA-binding motifs. Sox2 is able to directly interact with RNA. It recognizes G/C-rich RNA via a 60-amino-acid RNA binding motif (RBM) C-terminally juxtaposed to the HMG box (amino acids 120–180) [9]. The HMG box also has RNA-binding activity, but in a non-sequence specific manner [9,48]. In the presence of both DNA and RNA, the RBM preferably binds RNA, whereas the HMG box chooses DNA predominately. The RBM is suggested to regulate alternative splicing pre-mRNA of genes that play essential roles in cell fate determination by the association with splicing factors, such as Zcchc8, hnRNPK, and SPFQ, in the Sox2 interactome [9,49,50,51]. Deletion of the RBM alters splicing site selection of multiple Sox2-bound genes at exons that are G/C-rich around the 5’ splice site and ultimately disrupts the efficiency of somatic cell reprogramming, which suggests a role for Sox2 in the regulation of alternative splicing [9]. RNA-seq analysis indicated that the RBM controls the splicing patterns, but not the transcriptional levels, of certain genes [9], and confirmed the role of Sox2 in splicing regulation independent of the HMG box. This is in line with the fact that more than one third of the alternative regulators are TFs, including the HMG box-containing TFs [52]. Further, the RBM is also the region where Sox2 directly interacts with large intergenic noncoding RNAs (lincRNAs). Long intergenic noncoding RNA LincQ is specifically expressed in ESCs and is a central regulator of pluripotency. Sox2 binds to the exon 1 of LincQ via the RBM-containing segment, rather than the HMG box, and enrolls LincQ in transcriptional regulation of pluripotency-related genes [53]. The involvement of RNA in the Sox2 on-chromatin interaction network could be attributed to the complex interplay of Sox2, DNA, and RNA. In vitro RNA fishing assays indicate that Sox2 employs the HMG box and RBM to associate with DNA and RNA simultaneously [9]. It is possible that the RBM interacts with certain regions of pre-mRNA and affects exon selection co-transcriptionally, or it binds to noncoding RNA and changes the transcriptional properties of Sox2 while the HMG domain is in association with cis-elements on chromatin. 

Sequence-specific DNA-binding TFs can also alter mRNA isoform production by regulating transcription elongation rates [54]. Sox2 was found in the interactome with the positive transcription elongation factor b (P-TEFb), a heterodimer of Cyclin T1/T2, and cyclin-dependent kinase 9 (Cdk9), in Schwann cells [55]. The C-terminal domain (amino acids 180–319) of Sox2 interacts with Cyclin T1 spanning amino acids 472–505 and increases the elongation promoting activity of P-TEFb. The 7SK RNA is a small nuclear RNA that regulates transcription at regulatory regions. It forms a small nuclear ribonucleoprotein (snRNP) complex together with P-TEFb-associated protein HEXIM1 or HEXIM2 and inactivates P-TEFb [56]. Sox2 interacts with 7SK RNA in the nucleoplasm of mouse embryonic stem cells (mESCs) [57]. However, the Sox2 and 7SK RNA-involved complex does not function in the regulation of chromatin-based recruitment [57]. Alternatively, it may control transcription elongation in the context of P-TEFb and thereby influence pre-mRNA splicing.

### 2.2. Sry

Testis-determining factor SRY/Sry (sex-determining region Y), the gene of which is located on the Y chromosome, contains a conserved HMG box between species, but variant in other regions (Figure 1). It appears that the HMG box is essential for the Sry function. Indeed, nonsense and missense mutations in the HMG box of Sry leading to hermaphroditism or gonadal dysgenesis in human [58]. Knockout of the HMG box leads to XY sex reversal in Landrace pigs [59]. Similarly, deletion of the HMG box results in the production of XY females [60,61]. In vitro oligonucleotide binding assay identified that Sry preferably binds at an A/TTAACAAT/A sequence on linear double-stranded DNA (dsDNA). However, Sry is a fairly atypical TF. It shows high binding affinity to nonspecific DNA, which may be a major reason why its targets in vivo were not elucidated more than a decade after it was first cloned. Many target genes of Sry have been identified in recent years, such as *Sox9*, *Tcf21*, and *Ntf3* in testicular development [62,63,64]; *SAGA-associated factor 29* (*SGF29*) in hepatocarcinogenesis [65]; and *Cbln4*, rat *renin* (*Ren*), *angiotensinogen* (*Agt*), *angiotensin-converting enzyme* (*Ace*) and *Ace2* in the nervous system and neuroendocrine regulation [66,67]. It is noteworthy that Ace2 advantages the cellular entry of the severe acute respiratory syndrome-associated coronavirus 2 (SARS-CoV-2) and is highly activated in the serum of men with hypertension and heart failure in comparison with women, implying that Sry may be a reason for the higher mortality of men than women due to SARS-CoV-2 [68,69]. 

The HMG domain of Sry can also bind to the Holliday-like four-way junction DNA and RNA substrates [70]. During the early stage of mammalian spliceosome assembly, the postulated Holliday-like structure is crucial for defining splice sites and is formed adjacent to the 5′ and 3′ splicing sites [71]. Thus, it is possible that Sry bends the RNA associated by spliceosome, which in turn favors the protein/RNA-RNA interactions that allow the rearrangement and transition during the assembly of spliceosome. Ohe et al. showed that Sry is concentrated in nuclear with splicing factors and blocking the splicing redistributes Sry into enlarged nuclear speckles, implying its involvement in splicing [8]. Depletion of Sox6 in HeLa cells is unable to construct active spliceosomal complexes and impairs splicing activity. In complement with Sry, which includes a similar HMG box, it can restore the splicing in vitro [8]. 

The mere presence of the HMG box cannot initiate and sustain testis development [72]. Mouse Sry contains a C-terminal transactivation domain endowed with a polyglutamine (ploy-Q) region. This region is necessary for testis determination [73]. A recent study identified two variants of Sry proteins, Sry-S and Sry-T, that contain different amino acids after the poly-Q region. In nature, Sry-T, but not Sry-S, is sufficient to perform sex reversal in XX mice, indicating the predominant role of the C-terminal domain in male sex determination [74]. Despite its essential function, this domain is defective in human Sry and even some mouse strains like the *M. m. domesticus* subspecies (Figure 1) [73,75]. Sequence similarity searches and protein modeling cannot identify confident homologies of the C-terminal transactivation domain of mouse Sry. One reason may be because 56% of its sequence is predicted to be disordered. Knowing that that disorder in proteins may facilitate binding to the partners, we speculate the unique C-terminal domain as a recruitment platform for splicing factors [76,77]. Variation in the C-terminus may disturb the spliceosome assembly and ultimately testicular failure. 

### 2.3. Sox6

Sox6 is a key TF in the development of cardiac and skeletal muscles during both embryogenesis and postnatal development. Deletion of *Sox6* in mice leads to death two weeks after giving birth due to atrioventricular heart block and cardiac and skeletal myopathies [78]. Sox6 can act as a target and mediator of feedback to its upstream regulators. Sox9 is required for the expression of Sox5 and Sox6. In cooperation with Sox5 and Sox6, Sox9 binds to and activates chondrocyte-specific enhancers in genes such as *Col2a1* and *Col11a2* and regulates chondrogenesis in mice [79]. Sox2 and Sox6 reciprocally regulate the expression of each other and prevent premature neuronal differentiation during neurogenesis [80]. 

Remarkably, SOX6, like Sry, is a co-factor of alternative splicing. Sox6 colocalizes with splicing factors such as U1–70K, U2AF65, and U small nuclear RNAs in a majority of the nuclear speckles as well as in the areas between speckles. Sox6 dynamically redistributed into enlarged nuclear speckles if blocking splicing in living cells [8]. A lack of Sox6 abolishes active spliceosomal complexes formation and replenishing with the HMG box of Sox6 can recover splicing in vitro [4]. Accordingly, in line with the function of intact Sry, the HMG box of SOX6 could be the central part responsible for bending the spliceosome-associated RNA and gathering splicing factors during co-transcriptional splicing. 

DAX-1 is a dominant-negative transcriptional regulator of steroid hormone production. It acts as an anti-testis protein by functioning antagonistically to Sry. Distinct from Sry, Sox6 has a long region composed of two distinctive coiled-coil domains next to the N-terminus of the HMG box. DAX-1 interacts with the front coiled-coil domain of SOX6 both in vitro and in rat primary cultured sertoli cells [10]. DAX-1 interacts with splicing factor U2AF65 and inhibits pre-mRNA splicing. Sox6, as well as Sry and Sox9, competitively binds to DAX-1, reduces the U2AF65-DAX-1 interaction, and thereby promotes splicing through inhibiting the formation of U2AF65-DAX-1 inhibitory complex [10]. 

### 2.4. Sox9

Sox9 plays a key role in the development of the skeleton and is especially important for the determination of sex during embryonic development. As described earlier, Sox9 shares many similar roles with Sry and Sox6. For example, Sox9 can functionally substitute Sry HMG domain and trigger testicular development in transgenic mice [81]; Sox9 can, acting as Sry and Sox6, reduce the U2AF65-DAX-1 interaction and facilitate U2AF65-related splicing activity [10]. 

Sox9 also has unique regulatory roles in alternative splicing. Sox9 regulates the splicing of the transcribed RNA independent of its transcription property. Notably, both the HMG box and C-terminal transactivation domain are involved in this process. Mutants in the HMG box (Sox9-W143R, -A158T, -H165T, -P170R) and the C-terminal transactivation domain (Sox9-Del400, -Del440, -Del485), which abrogate the transcription ability, do not significantly affect the Sox9-associated splicing [11]. Moreover, two site mutants in the HMG box (Sox9- P108L, A119V) and two C-terminal deletion mutants (MiniSox9, Sox9-Del485) decrease the RNA binding, suggesting that the HMG box and the C-terminus are necessary for the RNA binding of Sox9 [11]. However, the regulation of splicing is not achieved by its RNA-binding activity, as Sox9-A119V and Sox9-Del485, which barely bind RNA, do not affect splicing activity [11]. Therefore, Sox9 probably regulates splicing by recruiting splicing factors to the Sox9-bound chromatin regions. Paraspeckle regulatory protein 54-kDa nuclear RNA-binding protein (p54nrb) physically interacts with Sox9 in interchromatin granule clusters, which function as spots for splicing factor storage and modification [82]. Sox9 could control the alternative splicing of *Fgfr2* transcripts at the chromatin level without affecting the transcription, implying a possible connection between Sox9 binding to its target chromatin regions and the splicing machinery [11,83]. By using proximity ligation assay (PLA), Girardot et al. detected several Sox9-associated RNA-binding proteins in DLD-1 cells, such as Y14/RBM8A, Sam68/KHDRBS3, p54nrb/NONO, PSP1/PSPC1, and PSF/SFPQ [11]. Knockout of Y14, PSP1, or Sam68 shows a similar effect to the depletion of Sox9. This demonstrates that Sox9 regulates splicing by gathering a regulatory complex including Y14, PSP1, and Sam68 to certain targets. Y14 is a component of the core exon junction complex (EJC), which builds on a pre-mRNA at the junction of two connected exons during splicing [84]. Sox9 is suggested to be a part of messenger ribonucleoprotein (mRNP) complexes together with Y14 and may confer gene specificity to EJC and probably other splicing factors as well during alternative splicing [11]. 

Sox9 connects with RNP transcripts of lampbrush chromosomes (LBCs), in which the active transcription occurs, in *Xenopus* oocytes [85]. However, Sox9 does not interact with the chromosome axis of the LBCs as a TF [85]. It binds to the RNA matrix in an RNA-dependent manner together with proteins such as CELF1 and SR proteins, which are associated with the nascent RNA transcripts and participate in the regulation of alternative splicing [85,86,87]. These findings support the role of Sox9 in co-transcriptional splicing. 

## 3. Role of Sox4 in Translation Regulation

Sox4 is another key TF required for vertebrate tissue development and differentiation. It is overexpressed in many types of cancer cells. Sox4 interacts with p53, a pivotal tumor suppressor, through its HMG domain and stabilizes it by blocking Mdm2-mediated ubiquitination and degradation [88,89]. Sox4 suppresses p53-mediated transcription in hepatocellular carcinoma (HCC) but increases the transcriptional activity of p53 in human colon cancer cells (HCT116) [88,89]. Sox4 has also been reported to inhibit the cell growth of glioblastoma multiforme (GBM) by mediating G0/G1 cell cycle arrest via Akt-p53 pathway and therefore is considered to be a positive prognostic biomarker of primary GBMs [90]. These seemingly contradictory regulatory roles of Sox4 are believed to depend on dynamically changing and cell-context based molecular interactions. 

Pol III specifically aims to transcribe short, stable nonprotein-coding RNA transcripts, which play important roles in the modulation of transcription (7SK RNA, B2 RNA, Alu RNA), RNA processing (U6 snRNA, small nucleolar RNA snR52, RNase P RNA), and translation (all tRNAs, 5S rRNA) [88,89,90]. Pol III is composed of 17 subunits that function together with 3 GTFs: TFIIIA, TFIIIB, and TFIIIC. Three types of Pol III promoters exist in mammalian cells. Unlike with Pol I and Pol II, the promoter sequences of most Pol III -transcribed genes, including genes of tRNAs and 5S rRNA, localize in the transcribed regions. Transcription of tRNA genes is regulated by GTFs, TFIIIB, and TFIIIC and a negative regulator Maf1 [91]. TFIIIC triggers the transcription of tRNA genes by directly binding to two conserved intragenic elements, the A and B boxes, allowing the recruitment of TFIIIB to the upstream-of-transcription start site. TFIIIB, which consists of three subunits, the TATA-binding protein (TBP), TFIIB-related factor 1 (Brf1), and Bdp1, subsequently increases Pol III and initiates tRNA transcription [92]. Strikingly, in addition to acting as a Pol II-dependent TF, Sox4 was recently found to regulate the transcription of certain tRNAs in a Pol III-dependent manner (Table 1). Sox4 targets many genes functioning in signal transduction and DNA-templated transcription. However, it is not involved in the Pol III-mediated transcription by influencing the expression of genes encoding TFIIIB subunits and TFIIIC. ChIP-seq data revealed that approximately 20% of the binding peaks of Sox4 are linked to tRNA genes [91]. Sox4 can target all copies of an individual tRNA isodecoder gene in human glioblastoma cells, e.g., 8 genomic copies of tRNAi^Met^ (located in Chromosome (Chr) 1, Chr 6, and Chr 17) and 6 genomic copies of tRNA^Phe^ (located in Chr 6, Chr 11, Chr 12, Chr 13, and Chr 19) [91]. On the other hand, Sox4 also shows specificity of binding—it does not bind to any of the 23 genes encoding nuclear-encoded mitochondrial tRNAs (nmt-tRNA) [91]. It was verified that Sox4 directly binds to tRNA genes, blocks the recruitment of Pol III, and thereby represses their transcription. Initiator tRNAi^Met^ is one of the most significant targets of Sox4. All 8 genomic copies of its gene are regulated by Sox4 [91]. CRISPR/Cas9-mediated knockdown of tRNAi^Met^ inhibits glioblastoma cell proliferation and the replenishing of ectopic tRNAi^Met^ partly rescues the Sox4-mediated proliferation inhibition [91]. 

p53, a partner of Sox4, is also believed to act as a general repressor of Pol III-mediated transcription. Endogenous cellular p53 interacts with TFIIIB and thereby severely impedes the function of Pol III. Since the function of p53 is retarded in the majority of cancers, e.g., HCC, it is probable that Pol III will be released from repression and induce the loss of growth control during the tumor’s development [92,93,94]. The similar modes of action suggest a possible coordination between Sox4 and p53 in Pol III regulation. 

## 4. Perspectives

Gene expression is primarily regulated at the level of transcription, largely as a consequence of the binding of TFs to specific sites on DNA. The function of TFs is to turn genes ‘on’ and ‘off’ in order to ensure that they are expressed to adapt to environmental changes throughout the life of the cells. Emerging studies have shown that TFs, such as Sox TFs, can control gene expression both at the step of transcription initiation and also during co-transcriptional pre-mRNA splicing by altering transcription elongation rates and/or influencing the interaction between splicing factors and pre-mRNA, in concert with the fact that more than one-third of the alternative splicing regulators are TFs [52]. More recently, diverting attention from the peculiar mode of DNA binding, researchers have found that many Sox members including Sox2, Sry, Sox6, and Sox9 fine-tune the coordination of transcription and splicing, suggesting that the final transcripts are decided not only by which genes the Sox TFs choose to turn ‘on’ but also how the Sox TFs trim concurrently or subsequently. Surprisingly, Sox TF can also regulate gene expression at the translational level. Sox4 gets into translational control by disrupting the recruitment of Pol III transcription machinery to tRNA genes and suppressing their expression [91]. Of particular note is that most of the gene regulatory roles of Sox TFs beyond transcription initiation are conducted by the HMG domain. Nonetheless, the non-HMG sequences confer a unique mode of action to individual Sox TFs. For example, the RBM of Sox2 binds to the G/C-rich regions of pre-mRNA and modulates the exon selection together with splicing factors; the first coiled-coil domain of Sox6 interacts with DAX-1 and causes an antagonistic effect in pre-mRNA splicing [9,10]. It is thus of interest to study the sophisticated network of regulatory events undertaken by Sox TFs.

## Figures and Tables

**Figure 1 ijms-22-05949-f001:**
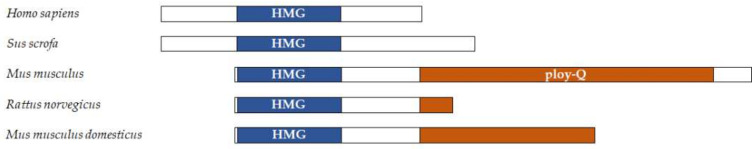
Comparison of structures of Sry protein from various mammalian species. Shown are the conserved HMG box (HMG) (in blue) and the polyglutamine (ploy-Q) region (in orange).

**Table 1 ijms-22-05949-t001:** Sox proteins and their functional domains.

.	Function	Functional Domain	Pol II-Dependent	Pol III-Dependent
SOX2	1. Regulating alternative splicing 2. Enroll non-coding RNA in transcriptional regulation 3. Control transcription elongation rates	RBM domain	Yes	
SRY	Regulating alternative splicing	HMG box C-terminal domain	Yes	
SOX6	Regulating alternative splicing	HMG box The front coiled-coil domain	Yes	
SOX9	Regulating alternative splicing	HMG box C-terminal domain	Yes	
SOX4	Regulating tRNA transcription	HMG box		Yes

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
