# Peer review of "Alternate Roles of Sox Transcription Factors beyond Transcription Initiation"

_ijms, 2021, doi:10.3390/ijms22115949_

Round 1

Reviewer 1 Report

Zhang and Hou describe the role of Sry and Sox transcription factors beyond DNA-binding functions of transcriptional activation/repression.  It is important to note transcription factors that facilitate pre-mRNA processing, isoform selection, and translational regulation of target genes.  An area not discussed in the review is chromatin accessibility and regulation, which is a critical function of pioneer transcription factors, such as Sox2 and Sox9.  Nevertheless, the review highlights the coordinated activity of transcription factors in target gene regulation beyond simply influencing expression levels.   Specific comments below in order of appearance in the manuscript. 

Introduction

  • The second paragraph of the introduction should have more citations of primary articles.
  • Line 22-23 - "Though can be studied in isolation in vitro, it becomes clear that each individual level is intimately linked within cells." - vague sentence, should be clarified. 

  • Line 25 - "key role in gene..."

  • Line 38 – are the authors referring to homo- and heterodimers with other Sox proteins or other cofactors. It could be worth clarifying here.

Roles of Sox TFs in splicing

  • Line 69 typo – should read TFIID
  • Line 135 -139 – the ideas do not come across well. I would reword this section with a few clear points.  There is also a typo in line 136: ‘evens’ should be ‘events’
  • Line 175-175 – three typos
    • ‘been’ should be eliminated
    • ‘more and more’ is better described as ‘Many’
    • ‘identified in recent years
  • Line 219 – typo, ‘In cooperation with’ instead of ‘cooperated’
  • Line 229 – typo, no “the” before splicing
  • Line 234-236 – would re-word sentence; perhaps start with “Distinct from…”
  • Line 243 – As described ‘earlier’ instead of ‘in the front parts’
  • Line 254 – can the authors provide more insight? If the splicing regulation of Sox proteins is independent of RNA-binding, then this suggests recruitment of splicing factors to SOX9-bound DNA?  Is there evidence of this activity?  Further elaboration would be useful for the reader.

Roles of Sox4 in translational regulation

  • Line 294 – typo; omit “many”

Reviewer 2 Report

In this manuscript Zhang et al. provide a literature review of how Sox transcription factors function in co-transcriptional pre-mRNA splicing by altering transcription elongation rates and by influencing interaction between splicing factors. The authors focus their discussion on Sox2, Sry, Sox6, Sox9 and Sox4 and provide a well encompassed review of the available data describing the roles of these Sox proteins in mRNA splicing. Overall, the discussion of this topic is thorough and well written. However, a few corrections are recommended to increase the impact of the manuscript.

  1. A general proofreading to fix the grammar style is recommended.
  2. There are some spelling errors that should also be proofread through the manuscript.
  3. It would be greatly impactful to include a figure showing the roles of Sox2, Sry, Sox6, Sox9 and Sox4 that are discussed in the text. Perhaps in a flow chart style. This would greatly strengthen the manuscript.
  4. Alternatively, instead of a figure a table may also be very helpful.
  5. Line 278-290: This information in this paragraph feels out of context and does not address any role of Sox4. This information should be moved to subsequent paragraph when mentioning the role of Sox4 in Pol III-dependent transcription.
  6. Lines 310-311: ‘SOX4 also….SOX4 [91].’ Please clarify the meaning and correct sentence structure.
